mSystems

# Average nucleotide identity-based *Staphylococcus aureus* strain grouping allows identification of strain-specific genes in the pangenome

Vishnu Raghuram,[1] Robert A. Petit III,[2] Zach Karol,[3] Rohan Mehta,[3] Daniel B. Weissman,[3] Timothy D. Read[2]

ABSTRACT  *Staphylococcus aureus* causes both hospital- and community-acquired infections in humans worldwide. Due to the high incidence of infection, *S. aureus* is also one of the most sampled and sequenced pathogens today, providing an outstanding resource to understand variation at the bacterial subspecies level. We processed and downsampled 83,383 public *S. aureus* Illumina whole-genome shotgun sequences and 1,263 complete genomes to produce 7,954 representative substrains. Pairwise comparison of average nucleotide identity revealed a natural boundary of 99.5% that could be used to define 145 distinct strains within the species. We found that intermediate frequency genes in the pangenome (present in 10%–95% of genomes) could be divided into those closely linked to strain background ("strain-concentrated") and those highly variable within strains ("strain-diffuse"). Non-core genes had different patterns of chromosome location. Notably, strain-diffuse genes were associated with prophages; strain-concentrated genes were associated with the vSaβ genome island and rare genes (<10% frequency) concentrated near the origin of replication. Antibiotic resistance genes were enriched in the strain-diffuse class, while virulence genes were distributed between strain-diffuse, strain-concentrated, core, and rare classes. This study shows how different patterns of gene movement help create strains as distinct subspecies entities and provide insight into the diverse histories of important *S. aureus* functions.

IMPORTANCE  We analyzed the genomic diversity of *Staphylococcus aureus*, a globally prevalent bacterial species that causes serious infections in humans. Our goal was to build a genetic picture of the different strains of *S. aureus* and which genes may be associated with them. We reprocessed >84,000 genomes and subsampled to remove redundancy. We found that individual samples sharing >99.5% of their genome could be grouped into strains. We also showed that a portion of genes that are present in intermediate frequency in the species are strongly associated with some strains but completely absent from others, suggesting a role in strain specificity. This work lays the foundation for understanding individual gene histories of the *S. aureus* species and also outlines strategies for processing large bacterial genomic data sets.

KEYWORDS  genomics, pangenome, *Staphylococcus aureus*, antimicrobial resistance

$S$taphylococcus aureus is a ubiquitous human pathogen capable of causing numerous disease manifestations, including more than 100,000 bloodstream infections in 2017 in the USA alone (1). *S. aureus* genomes typically have a ~2.8-Mb chromosome and zero to a few plasmids. Like other bacterial pathogens, its success at responding to pathogenic niches comes from adaptations in the "core" portion of the genome and in non-core genes that form the extended species genome, or "pangenome" (2). Non-core genes form part of the extensive genetic repertoire for evading the immune response

Editor Juliette Hayer, Institut de Recherche pour le Developpement, Montpellier, France

Address correspondence to Timothy D. Read, tread@emory.edu.

The authors declare no conflict of interest.

See the funding table on p. 17.

and damaging the host and have allowed *S. aureus* to survive treatment with various antibiotics developed since the middle of the 20th century (3–6).

Microbiologists have long known that there are consistent differences in phenotypes between taxonomic groups below the species level in *S. aureus*. Different "strains" have been shown to be more likely to cause specific disease etiologies than others. Examples are multi-locus sequence type (MLST) ST582, which is associated with scalded skin syndrome (7) and livestock-associated CC97 infections (8). Among other phenotypes, strains also show different propensities to acquire drug resistance genes and high or low levels of toxin production and can produce different spectra of mutations when under strong selection (9–12). Understanding the genetic basis of strain specificity therefore offers potential insight into many mechanisms that define *S. aureus* pathology. Interest in strain specificity has also been prompted by attempts to use shotgun metagenomic data to define environmental conditions that separate different genotypes with species (13, 14). However, the cardinal problem with these approaches is that there is no generally accepted bacterial strain definition appropriate for the genomic era. Instead, the term "strain" has been used loosely to apply to different levels of subspecies variation.

The aims of this work were to seek a consistent definition of a *S. aureus* strain that could be applied to genomic and ultimately metagenomic data, to understand which portions of the non-core genome were strain associated, and to survey the extent of strain variation in the public data. We used an approach based on an earlier workflow (12) where we reprocessed all extant public Illumina whole-genome shotgun (WGS) data. Here, we refined the strategy by implementing stringent steps to filter WGS potentially contaminated with other bacterial contigs and *S. aureus* mixtures. We also included high-quality complete genomes and dereplicated the final data set to remove highly similar sequences. Critically, we opted to define relationships between genomes based on average nucleotide identity (ANI), rather than relying on the traditional clonal complex (CC) and sequence type (ST) designations of multi-locus sequence typing.

## RESULTS

### ANI threshold of 99.5% defines 145 *S. aureus* strains from a large public genome data set

To obtain a comprehensive view of *S. aureus* genetic diversity, we first examined all 83,383 whole-genome data sets available on the National Center for Biotechnology Information website in September 2022 for data quality and created a curated data set with 58,034 genomes. Then, we filtered to reduce redundancy and selected 7,954 high-quality "substrains" that represented the overall diversity (Fig. 1; Fig. S1A; Materials and Methods).

The 7,954 representative substrains were used to create a species pangenome (the "7954-set") using the PIRATE software (15) based on a minimum of 50% protein sequence identity. A total of 9,533 distinct gene families were identified (we use the shortened "genes" to refer to these gene families in this article). Of these genes 2,008 (21.1%) were considered core (found in >95% of the genomes); 71.3% (6,794) were rare (<10% of genomes); and 7.7% (731) were intermediate between core and rare. Ninety percent of the genes were in single copy (Fig. S2).

When pairwise ANI between substrains was plotted as a histogram, we observed three major ANI peaks (Fig. 2A). We interpreted the left peak (smallest average nucleotide distances) as intrastrain distance and the second and third as between-strain distances within the two major *S. aureus* clades (16) and between the clades, respectively. The threshold for intrastrain relatedness appeared to be at, or very near to, 99.5%, identical to a value suggested by Rodriguez-R et al. to separate strains across 330 bacterial species (17). When we used 99.5% as a threshold for clustering, we obtained 145 groups of genomes that we termed strains and marked each with a suffix, "99.5_i," where i denoted the unique strain number of one of the 145 strains. All strain clusters had median within-cluster ANI of >99.7 (Fig. S1B). Both gene discovery rate and lineage discovery rate were

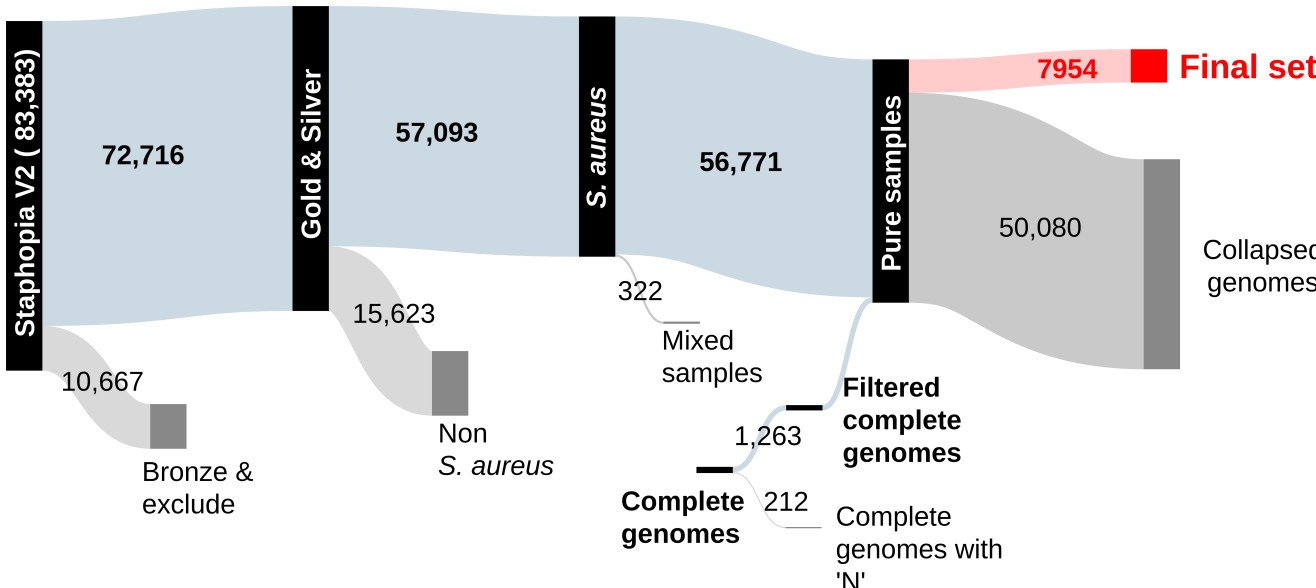

**FIG 1** Sankey diagram showing the fate of 83,383 *S. aureus* whole-genome shotgun data sets and 1,475 complete genomes through processing and filtering.

improved by dereplicating the initial curated 58,034 genomes compared to using a random set (Fig. S1C and D).

Of the 10 CCs defined by the *S. aureus* PubMLST site (18), 7 were split across different strains at the 99.5% clustering threshold (Fig. 2B). The most extreme example was CC1, where pairs of substrains differed by as much as 1.4% ANI (Fig. 2C), and the clonal complex was ultimately split across 14 strains. Across all strains, we found that >99.9% of the genomes in the same strain had the same *agrD* specificity allele (1–4) of the *agr* quorum-sensing system (Fig. 2D). [The one exception was strain PS/BAC/317/16/W (GCF_018093225.1) (19), the single *agr* group 2 genome in 4,469 CC30 genomes.] This result confirmed an earlier genome-based screen (20) showing that the *agr* type is strongly strain specific in *S. aureus*.

We noted that there was a "bump" of pairwise distances (~99.5% to 99.1% ANI) in the otherwise clear gap between within-strain and between-strain comparisons (Fig. 2A). When we clustered substrains at 99.1% core genome ANI, we found that 30 99.5%-defined strains merged together to form 115 putative strains. One of the merged strains comprised genomes of S99.5_2 and S99.5_27, both largely mapped to CC8. The S99.5_27 strain consisted of ST239, which is known to have been created by the recombination of a large portion of a CC30 genome with a CC8 background (21, 22). The other nine sets of merged strains consisted of a small number of genomes. For two of the merged strains, we had a complete genome which we used to align 10,000-bp sliding windows against a genome from the same strain at 99.5% ANI and one from a different strain that was merged at 99.9% ANI. These were strains S99.5_33 and S99.5_4 (both mapped to CC45) and S99.5_7 and S99.5_111 (CC15), each pair merged into one strain using ANI 99.1% thresholds. Neither analysis revealed the clear pattern of large-scale genome replacement seen in ST239. All but 3 STs out of 1,706 mapped only to one strain. The exceptions were two CC45 STs that mapped to three different strains (S99.5_33, S99.5_4, and S99.5_57) and one CC15 ST that mapped to two different strains (S99.5_7 and S99.5_111).

## Intermediate frequency genes in the pangenome can be divided into strain concentrated and strain diffuse

We wanted to know what proportion of the *S. aureus* non-core genes were strongly linked to strain background, in the same manner as *agr* type. We adapted the commonly

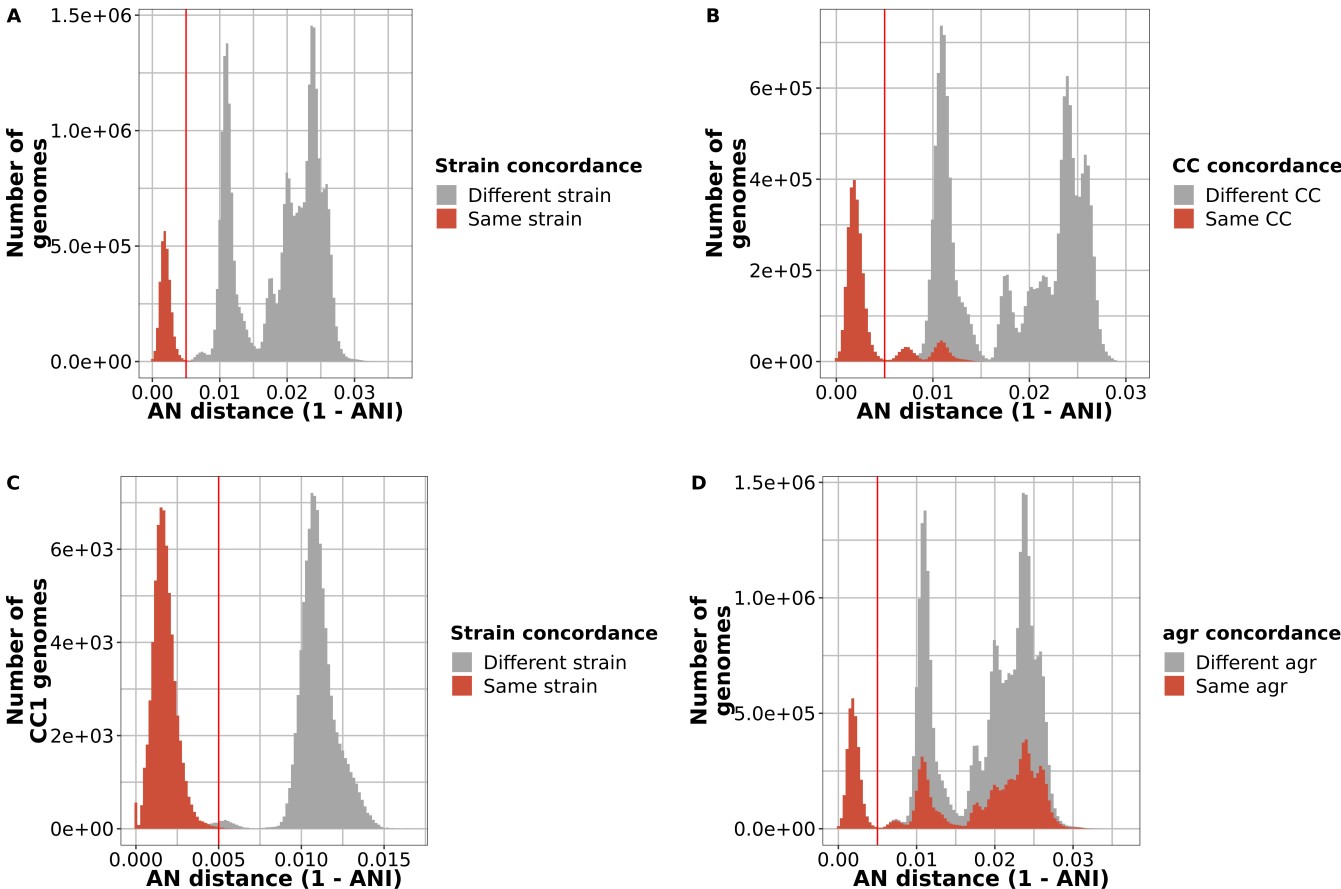

**FIG 2** An average nucleotide identity of >99.5% defines the strain boundary of *S. aureus*. For our data set of 7,954 substrains, all-vs-all pairwise average nucleotide (AN) distances were plotted as a histogram. (A) Sample pairs less than 0.005 AN distance apart (i.e., greater than 99.5% ANI) were grouped as a strain. (B) Strains and clonal complex (CC) designations do not exactly overlap. The pairwise AN distance histogram was colored by whether the genomes were in the same CC. (C) CC1 genomes are in different strains. AN distances of genomes assigned to CC1 showing that there are within- and between-strain distances. (D) Genomes in the same strain have the same *agr* group. The pairwise AN distance histogram was colored by whether the genomes were in the same *agr* group.

used genetic statistic, also known as fixation index ($F_{ST}$), as a measure of segregation of a gene between different strains (23). A $F_{ST}$ of 0 indicated a gene that displays no genetic segregation; i.e., it was indiscriminately found across different strains. In contrast, a $F_{ST}$ of 1 indicated perfect genetic segregation, with the gene limited to all members of a group of strains. Rare and core genes were constrained in their distribution and had uninformative $F_{ST}$ scores around 0. Therefore, we focused our analysis on intermediate gene families.

Strikingly, the $F_{ST}$ statistic across intermediate genes showed a distinct bimodal distribution (Fig. 3A). This pattern disappeared when the strain labels were randomly mixed and $F_{ST}$ was recalculated (Fig. 3B), reverting to a normal distribution, showing that it was a feature of the specific population structure of *S. aureus* rather than an inherent property of the data. From this result, we divided intermediate genes into two groups based on a $F_{ST}$ threshold of 0.75. Those genes with high $F_{ST}$ [295 of 731 (40%) intermediate genes], which we termed strain concentrated, were strongly linked to strain backgrounds, while those with low $F_{ST}$ (strain diffuse) [436 of 731 (60%) intermediate genes] were more promiscuous, with respect the strain background. These patterns were illustrated using 10 *S. aureus* toxins with a range of $F_{ST}$ scores: leukocidins LukFS (Panton-Valentine leukocidin) and LukED, toxic shock syndrome toxin 1 (TSST), superantigen-like protein SSL8, and different types of staphylococcal enterotoxins (SEA, SEB, SEG, and SEU) (Fig. 4). Leukocidins comprise two proteins, the F component and

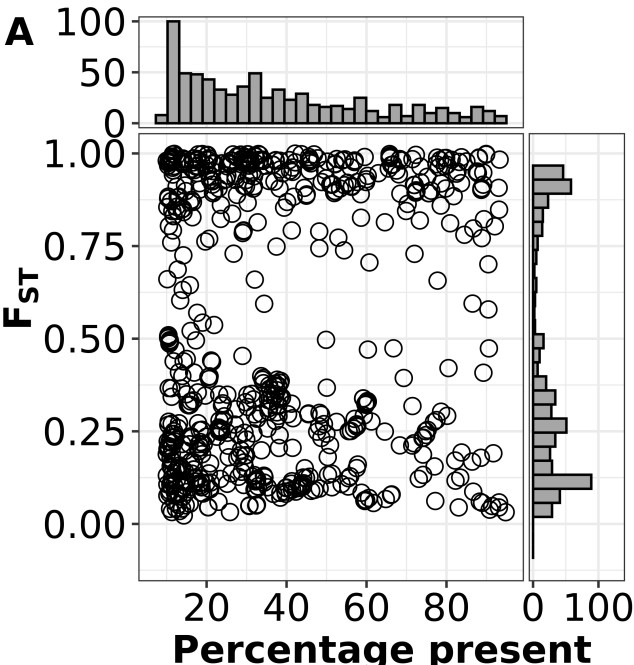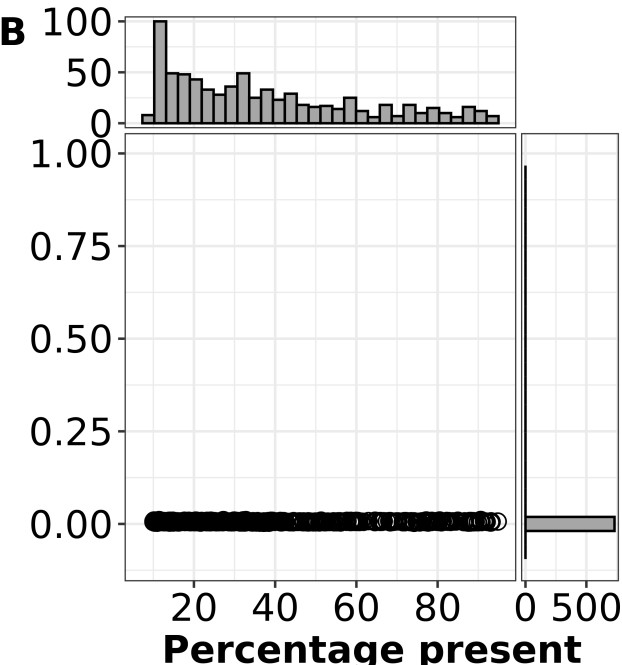

**FIG 3** Bimodal distribution of $F_{ST}$ for intermediate genes. Each circle represents an individual intermediate gene from the 7,954-substrain pangenome. Percentage prevalence on the *x*-axis is the percentage of genomes the gene is found in. $F_{ST}$ or "fixation index" is on the *y*-axis. (A) $F_{ST}$ scores calculated for each intermediate gene with 99.5% ANI-based clustering. (B) As a control, $F_{ST}$ scores were calculated for each intermediate gene when clusters were randomly assigned.

the S component, both acting synergistically to form pores in host-cell membranes (24). TSST, SEs, and SSL8 are superantigens or superantigen-like proteins, highly potent toxins that can elicit severe inflammatory responses and other immunomodulatory effects (25). The leukocidin LukFS, enterotoxins SEA and SEB, and TSST, showed high levels of gain and loss on the species tree typical of low $F_{ST}$. In contrast, the enterotoxins SEG and SEO, and leukocidin LukED, found together on genomic island vSaβ had high $F_{ST}$ (>0.9) and were either almost entirely present or absent in each strain background. For example, LukD was not present in any substrain of 60 of 145 (41%) strains but present in >80% of the substrains of 77 (53%) strains.

We also used $F_{ST}$ to test whether there was any association between the *agr* type of a strain and intermediate gene distribution but found no similar pattern (Fig. S3A).

The 7,954 representative substrains were distributed unevenly, with 58 strains having a single substrain and 15 strains having >100. This "unbalanced" sampling was an obstacle to visualizing gene abundance patterns. Genes that were present even in a low percent of the most numerous strains would still account for more substrains than the rarest strains. We created the "740-set," created by randomly sampling 20 shotgun assembled substrains from the top 37 most populous strains to make a more balanced sampling of *S. aureus* (Materials and Methods). The 740-set had similar numbers of core and intermediate genes (2,139 and 739, respectively) to the 7954-set but fewer rare genes (2,687), the latter expected to increase with the number of genomes sampled in a species. The $F_{ST}$ distribution of the 740-set to the original pangenome was almost identical (Fig. S3B).

When we plotted the number of strains each gene was found in, given the numbers of genomes, we saw two distinct patterns. The strain-concentrated genes were close to the minimum possible number of strains for a given gene (solid black line), while the strain-diffuse genes were more similar to the shape of a random assortment of strains (asymptotic exponential distribution, dashed black line) (Fig. 5A). Strain-diffuse genes

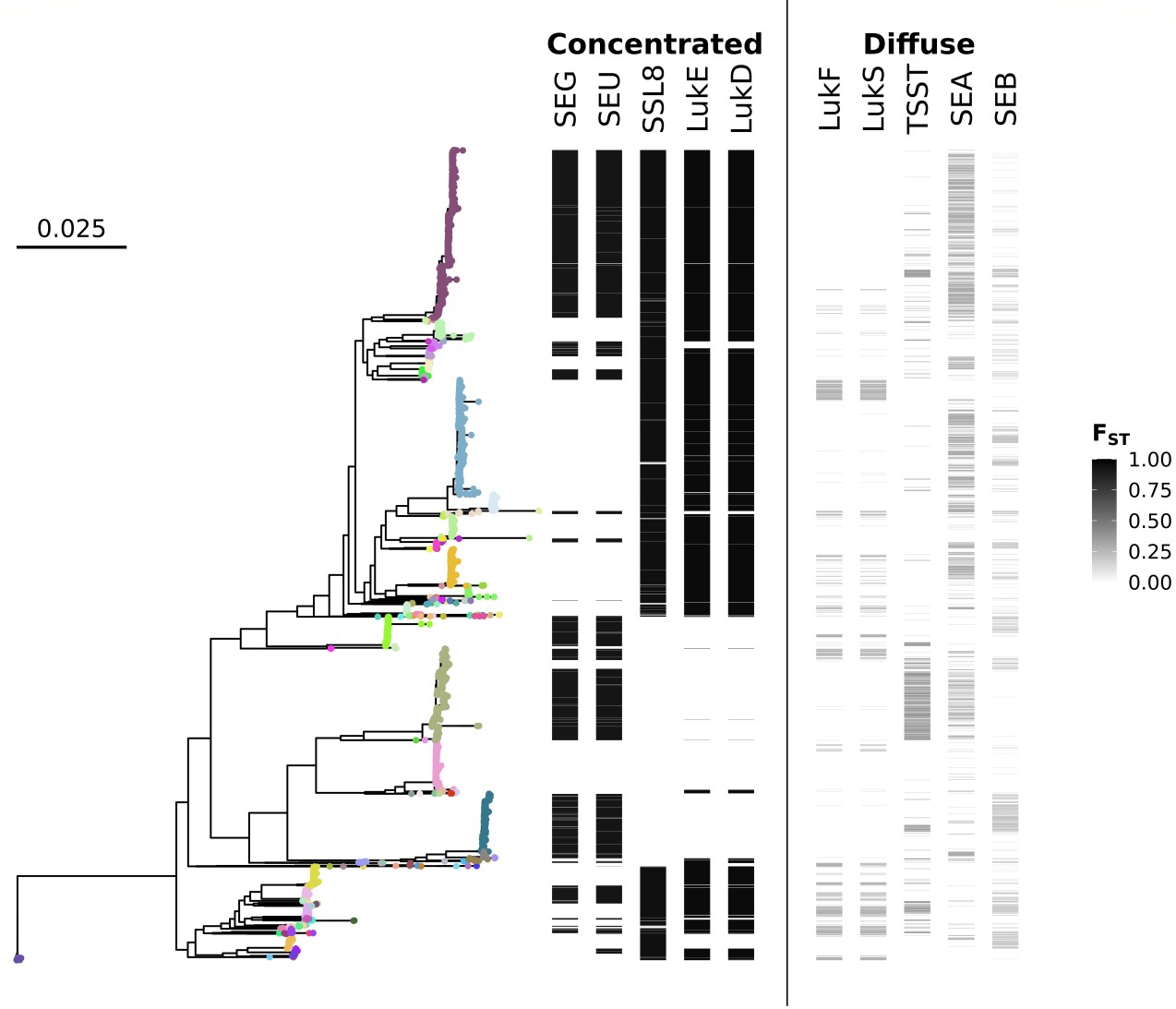

**FIG 4** Strain-group specificity and co-occurrence of specific staphylococcal toxins. Core genome phylogeny of the 7954-set. Heatmap on right shows presence absence and $F_{ST}$ of specific staphylococcal toxins: Panton-Valentine leukocidin (LukF and LukS), toxic shock syndrome toxin (TSST), and staphylococcal enterotoxins types A, B, G, and U (SEA, SEB, SEG, and SEU), superantigen-like protein (SSL8), and leukocidin ED (LukE and LukD). The colors of the whole-genome phylogeny are based on strain assignments.

were present in markedly more strains at a given prevalence than strain-concentrated genes.

Figure 3 and 4 depict a pattern where strain-diffuse genes appeared to undergo gain and loss on the phylogenetic tree at a higher rate than strain-concentrated genes . Based on the results from HomoplasyFinder (26) on the core gene phylogeny of the 740-set, we found this pattern was consistent across all intermediate genes (Fig. 5B). Strain-concentrated genes mostly had fewer than 30 minimum predicted state changes on the tree, and there was no trend in increase of this number with prevalence. Strain-diffuse genes had a higher rate of character state change, which rose with prevalence initially but fell with the most common genes, probably due to saturation of available state changes.

Because of the relatively slower rate of gene gains and losses, the strain-concentrated genes contributed more to characteristic strain-specific differences in gene content than strain-diffuse genes. This could be effectively visualized using t-distributed stochastic neighbor embedding (t-SNE, Fig. 6). When strain-concentrated genes' presence/absence

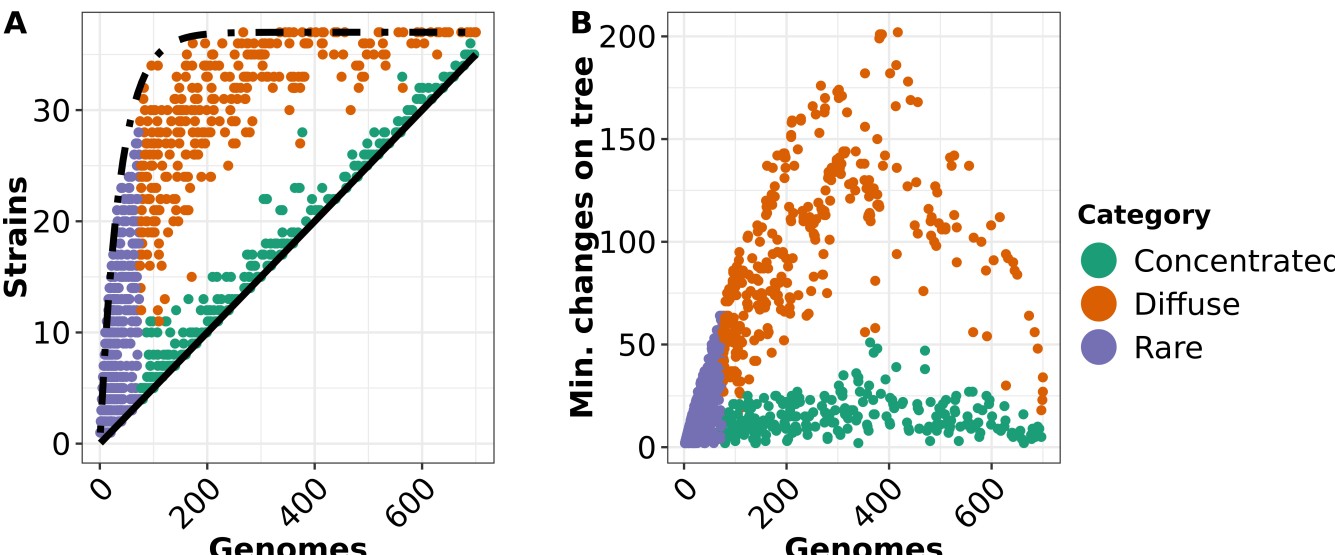

**FIG 5** Relationship between gene prevalence, number of strains, and homoplasy for non-core genes. Each dot represents a non-core gene in the 740-set pangenome. Purple denotes rare genes; green denotes concentrated genes; and brown denotes diffuse genes. (A) The relationship between overall prevalence (number of genomes out of 740) and number of strains (out of 37) each gene is found in is shown. The curves for the theoretical minimum number of strains for a given number of genomes ($x$/20) are shown in solid black, and the extreme random distribution ($37 \times (1-\exp(-x/37))$) is shown in dashed black. (B) The relationship between prevalence of estimated number of changes on the species tree calculated by HomoplasyFinder (26).

was used as input for t-SNE, the genomes that comprised individual strains were resolved into distinct spatial units (Fig. 6C). However, there was no similar pattern when strain-diffuse genes were used (Fig. 6B). Rare genes produced an intermediate result, with some distinctive strains and some areas of the plot with mixtures of strains (Fig. 6A). When all non-core genes were used, the strains could be readily distinguished, indicating that for the t-SNE approach, the strain-specific structure of strain-concentrated and rare gene content was dominant to the non-strain-specific strain-diffuse genes (Fig. 6D). We also visualized the effect of the different classes of non-core gene is a way that was independent of strain classification: plotting the gene content similarity (represented by hamming distance) of each pair of genomes against the patristic distance on the core gene phylogeny (Fig. S4). The rare and strain-diffuse genes had greater numbers of gene differences between strains very closely related to each other (Patristic distance <0.005), but the rate of growth of the distance in strain-concentrated genes over larger distances on the phylogeny was greater.

We suspected that the underlying differences between the two groups of genes were due to strain-concentrated genes being primarily located on the chromosome and primarily spread between strains by homologous recombination, whereas strain-diffuse genes were on mobile elements such as prophages, plasmids and integrative conjugative elements that would be located more frequently on non-chromosomal contigs. This was supported by the rate of linkage to single copy highly conserved core genes (defined as whether the gene was found to be on the same contig) was much lower in strain-diffuse genes (65.5%) than strain-concentrated (86.5%). By comparison, the rates for rare genes were 61.5% and those or for randomly selected genes were 93.5%. We used the geNomad software and database of mobile element genes (27) to see if there were different distributions in the different classes of genes in the pangenome. While differences between the classes were mostly statistically significant at $P < 0.05$ in pairwise Tukey's tests (Fig. S5), the differences in mean scores were mostly quite small, probably reflecting the relatively small size of the *S. aureus* training set for the software compared to our large pangenome sampling. The strain-diffuse genes had the most distinctive signal, having the lowest mean scores for "chromosome" and "plasmid" and

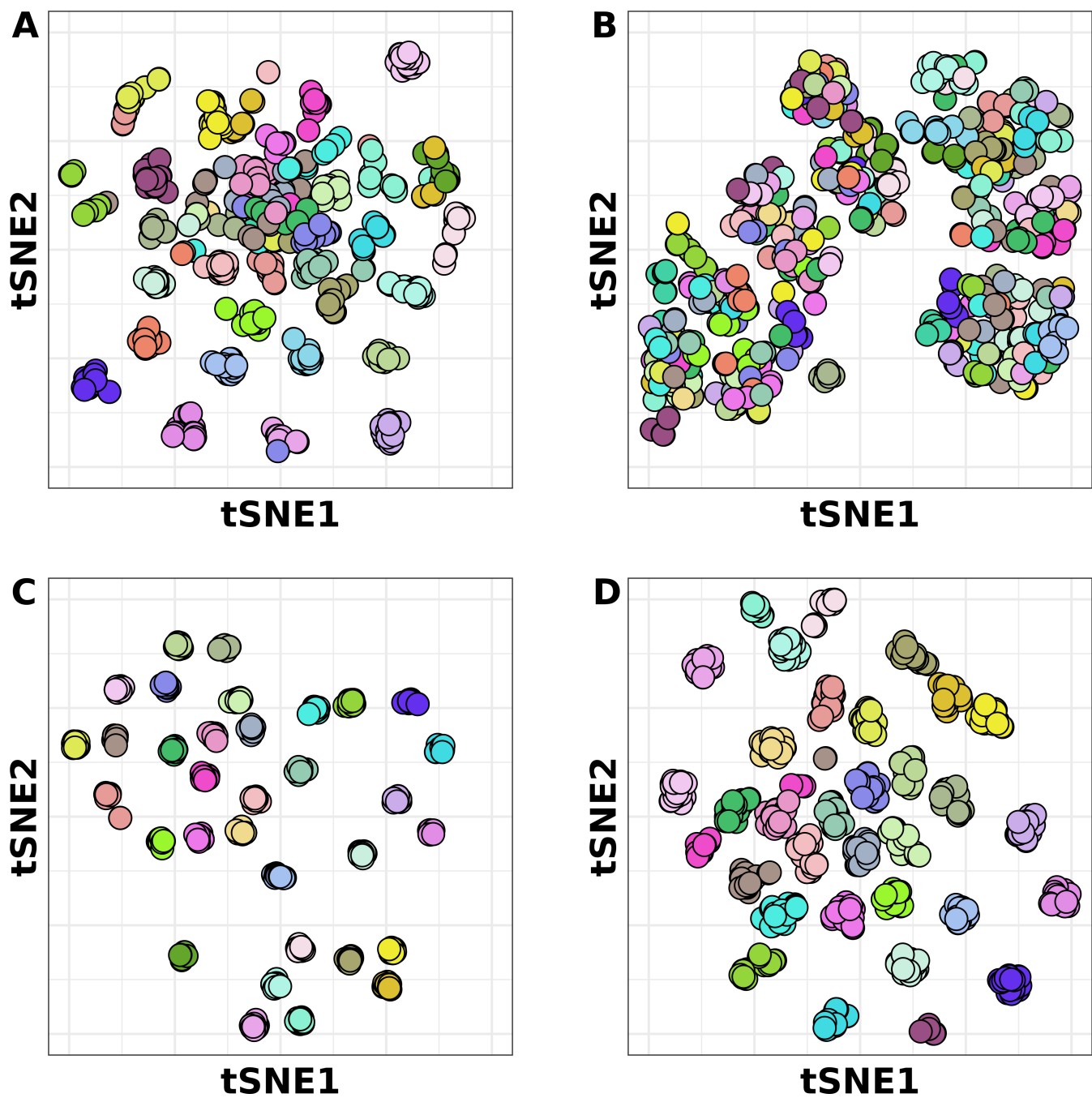

**FIG 6** t-SNE analysis of 740-seq differentiated by non-core gene sets. Each dot represents one of the genomes of the 740-set colored by its strain membership. Different sets of non-core genes were used as input for the t-SNE: (A) only rare, (B) only strain-diffuse, (C) only strain-concentrated, and (D) all non-core.

highest for "virus." This result corroborated the association of strain-diffuse genes with prophage regions of the genome.

We noted that the intermediate genes had a lower median clustering threshold than the rare or core genes [the PIRATE software uses iterative thresholds at increasing stringency to find the final clustering threshold for a gene (15)]. To ensure the patterns seen were not an artifact of lower clustering, we ran the 740-set pangenome with a minimum clustering threshold of 90% amino acid identity (which we called "740-set-90"). While the more stringent clustering split several rare and intermediate gene families (the 740-set-90 pangenome consisted of 4,490 rare, 982 intermediate, and 2,085 core),

the characteristic divergence in features between strain-concentrated and strain-diffuse genes did not change (Fig. S6). We also obtained similar results when the same analyses were run with the original 7,954-substrain pangenome, although the unbalanced nature of the collection (some strains had thousands of genomes, many only one) obscured the differences between strain concentrated and strain diffuse in regard to the relationship between strains each gene was detected in at different prevalence (Fig. S6A). The strain-concentrated genes though had many fewer predicted state changes on the phylogenetic tree (Fig. S6B).

## Different non-core gene classes cluster in specific regions of the *S. aureus* chromosome, with a strong tendency for rare genes to be near the origin of replication

We used two alternative methods to view the distribution of non-core genes on the *S. aureus* chromosome (Fig. 7; Fig. S7). In the first method, we plotted the start coordinate of genes from 337 complete chromosomes (Fig. 7A; Fig.S7). There was noise in the exact coordinates of individual genes, but overall, this method showed discrete peaks in the locations of rare, strain-concentrated, and diffuse genes. The second method was to link non-core genes from all 7,954 substrains to the nearest core gene on the same contig (non-core genes on contigs without core genes were excluded). The gross patterns of distribution of the counts of non-core genes mapped to the nearest core gene coordinate (Fig. 7B) were similar to that in Fig. 7A. Differences between plots in the proportion of genes within each category at each genomic bin (*y*-axis) were probably due to a combination of the indirect measurement of gene position in the linked core gene method and the fact that the 7,954 substrains were more balanced reflection of *S. aureus* diversity than the 337 complete genomes.

Strain-diffuse and strain-concentrated genes had markedly distinct distributions on the chromosome and were mostly located as part of distinct clusters (Fig. 7). This could also be seen clearly in the individual chromosomes of six substrains chosen to represent both methicillin-resistant *S. aureus* and methicillin-sensitive *S. aureus* from three strains (Fig.S7). The vSaβ genome island was a notably strain-concentrated-rich gene cluster, while the vSaγ island, phiSa2, and phiSa3 prophages were rich in strain diffuse. The presence of strain-diffuse gene clusters was more variable between genomes than strain-concentrated clusters (Fig. S7). Some genetic elements (e.g., SCCmec, type VII secretion loci, and phiSa1) contained a relatively high proportion of both types of intermediate genes. Three regions of the chromosome relatively rich in strain-concentrated genes (at approximate coordinates 100,00–300,000, 1,250,000–1,500,000, and 2,500,000–2,800,000) did not correspond to known genetic elements , although the first region contained several genes involved in polysaccharide capsule synthesis.

The high number of rare gene genes in the 0–100,000 region (which includes the SCCmec cassette) was an outlier compared to other chromosomal regions (*P* value <2.2e-16, Grubbs one-tailed test) (Fig. 7; Fig. S7). This was the case in both MRSA and MSSA strains, suggesting that this region might be a hotspot for insertion of rare genes, possibly through plasmid integration, rather than being specifically linked to SCCmec.

## Functional differences in strain-concentrated and strain-diffuse genes

$F_{ST}$ and prevalence of intermediate gene families can provide insight into ongoing evolutionary processes in the species. This is illustrated by analysis of three classes of genes encoding antimicrobial resistance (AMR) phage defense and virulence determinants (Fig. 8). No AMR genes (30) were found to be in the strain-concentrated group but were either rare or strain diffuse [70 (82.4%) and 15 (17.6%), respectively] . This result follows from the recent introduction of many AMR genes into *S. aureus* on mobile genetic elements and their frequent gains and losses below the strain level (31). The absence of fixation within strains also suggested possible loss of mobile elements in the absence of antibiotic selection. Genes associated with protection from phage infection in the defense-finder database (32) were mostly low prevalence [69 of 80 (86.3%) were rare,

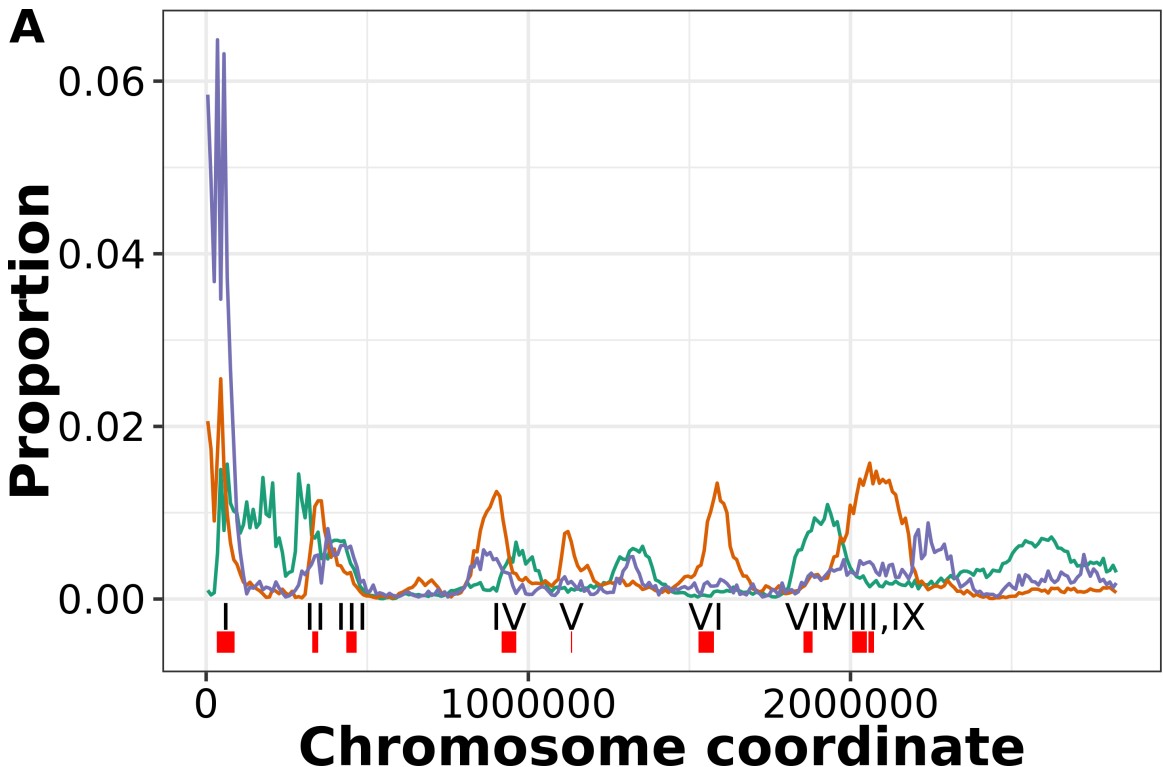

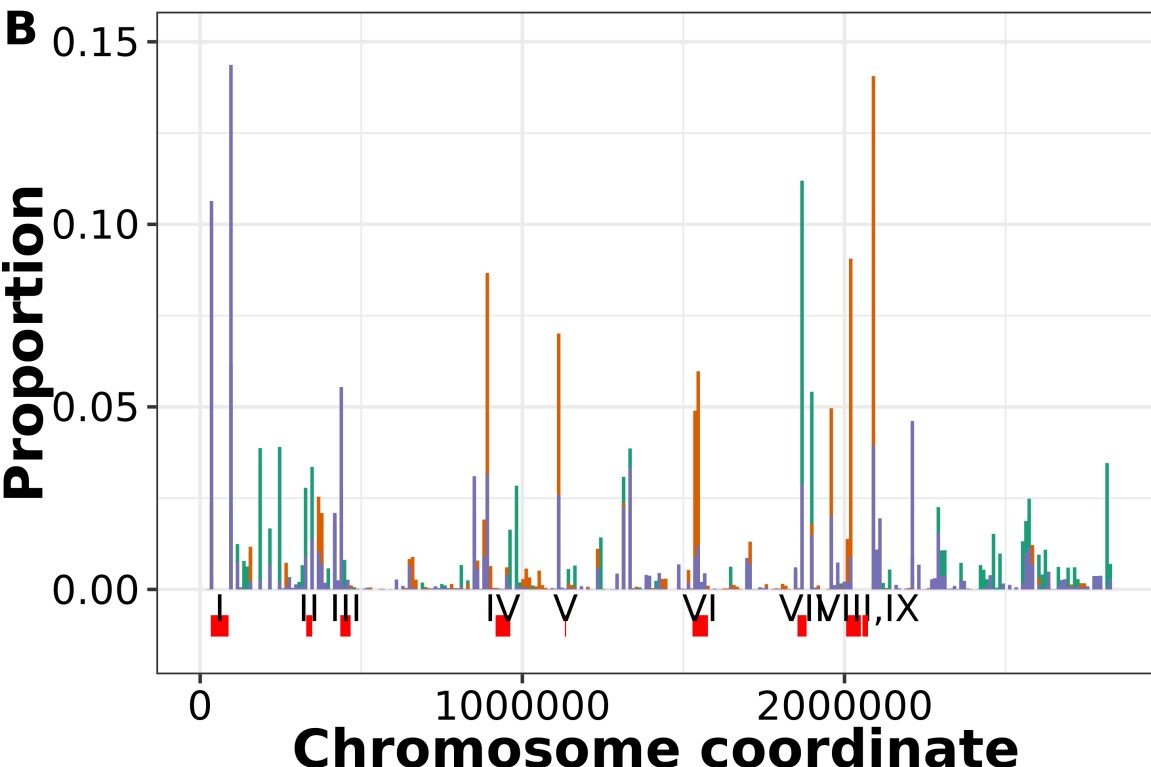

**FIG 7** Distribution of different categories on non-core genes on the *S. aureus* chromosome using two alternative methods. (A) Location based on 337 complete genome sequences. The start site for every gene in each category was obtained for 337 chromosomes. The totals were placed in 10,000-bp bins on the chromosome, and the proportion of the total for each class is plotted (i.e., the sum of the values of the 10,000 bins is 1). Purple denotes rare genes; green

**FIG 7** (Continued)

denotes strain-concentrated genes; and brown denotes strain-diffuse genes. (B) Location based on the nearest core gene. For all 7,954 substrains, the closest core gene on the same contig was determined. The *x*-axes are start sites for the core genes of genome N315 (GCA_000009645) (28). The values were binned and proportionalized as in panel A. For both panels A and B, the location of selected features is shown: I, SCCmec; II, type VII secretion system; III, vSaα; IV, phiSa1; V, vSaγ; VI, phiSa2; VII, vSaβ; VIII, phiSa3; IX, vSa4. N315 coordinates are based on Gill et al. (28) and Warne et al. (29), except phiSa2 and phiSa3, which are from Mu50 and MW2, respectively.

and 10 of 80 (9.1%) intermediate had prevalence <0.5]. The low prevalence may reflect diversifying selection caused by phage countermeasures. However, unlike AMR genes, the majority of intermediate genes in this class were strain concentrated, suggesting that defense from phage infection may help define *S. aureus* strains. Intermediate virulence genes [mostly toxins (33, 34)] in the AMRFinder+ database fell into two groups: one strain diffuse with low prevalence and the other strain concentrated with mostly higher prevalence. Strain-diffuse virulence genes were mostly associated with prophages and Sa-PIs, while strain-concentrated genes were associated with the vSaβ genome island. This partition suggested an as-yet unexplained complexity in the hierarchy of functions that make up the toxin profile of an individual substrain.

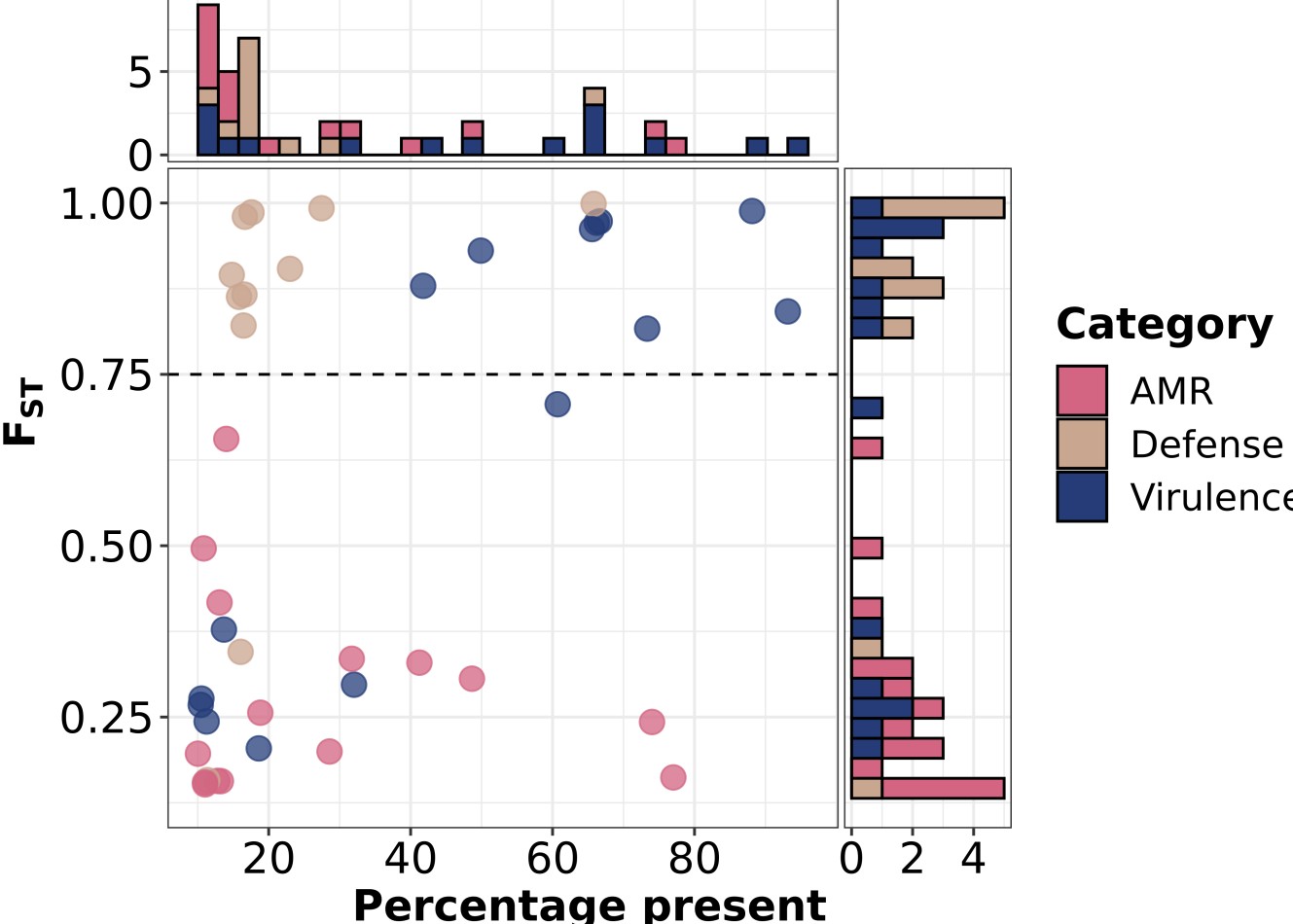

**FIG 8** Prevalence vs $F_{ST}$ for intermediate antimicrobial-resistance (AMR), virulence, and phage defense genes. AMR and virulence genes were identified using AMRFinder+ (30); phage defense genes were identified using defense-finder (32). The dashed horizontal line represents the boundary between strain diffuse and strain concentrated.

## DISCUSSION

In this study, we distilled a starting set of >84,000 *S. aureus* genome sequences to 145 strains using an ANI cutoff of 99.5%, which we found to be in a natural valley between clustered isolates. This threshold, or values close to it, has been reported in other studies as a bacterial subspecies boundary (17). A large number of *S. aureus* strains were rare [92 of 145 (63.4%) represented by one to two substrains]. While this could represent some aspect of the true distribution of strain abundances in the species, it could also be a function of uneven sampling of *S. aureus* genomes. There are large ascertainment biases in selection as most strains are from clinical settings in Western countries. It is probable that the number of strains will grow significantly in the future as we extend sampling.

The 145 representative genomes defined here could be used for assignment of a new genome to an existing strain using fastANI or similar software. This simple approach for strain assignment has the advantage of not needing a core phylogeny calculated that is inherent to tree-based clustering and may turn out to be similarly accurate owing to the population structure of the within- and between-strain differences in the species (Fig. 1). In some cases, we found that the current MLST-based CCs were split into more natural strain clusters by ANI. This is not surprising, as MLST schema was developed for PCR amplification and sequencing, before routine whole-genome sequencing was available, and the seven loci used for assignment only cover a small portion of the variation in the chromosome (35, 36). MLST, though useful for rapid strain typing, is outperformed by whole genome-based methods for lineage assignment (36, 37).

Several pangenome studies with *S. aureus* genomes have been performed for epidemiological investigations (38–43), vaccine candidate discovery (44, 45), and evolutionary phylogenomics (46–49). These produced a wide range of results, from a total pangenome size of 4,250–21,358 genes, with cores ranging from 890 to 2,700 genes (Table S1). The variability is a feature of the many factors that influence pangenome estimation, which can be classed into three main groups: sample collection, data quality, and bioinformatics approaches. In terms of the collection, more individual genomes of a species tend to produce a larger number of gene families (in an "open" pangenome) and smaller core (50). Similarly, the more genetic diversity within the species increases pangenome size. We used essentially all the genome data available in the public domain by Fall 2022 [although we ended up excluding several thousand experiments based on quality (Fig. 1)]. Therefore, this study probably has the largest and most diverse input of *S. aureus* set used to date. By reducing genome redundancy, we also mitigated some of the overcounting of highly sampled clones in the public databases. Ideally, all genomes for a pangenome should be of high quality and complete. However, we chose to include shotgun assembled genomes, which may contain a certain percentage of missing genes due to contig breaks, to maximize diversity. Using shotgun assemblies also allowed us to sample multiple genomes from a larger number of strains, which was important for characterizing strain-diffuse and strain-concentrated genes. By reprocessing the data from raw reads, we were able to filter out lower-quality data and have consistent assemblies (Fig. 1). In tests, we found that pangenomes based on our shotgun assemblies produce metrics similar to those estimated using only complete genomes, as evidenced by the 740-set, which was composed entirely of shotgun data. For most complete genomes, there is no matching raw read data available in public archives, so it is not possible to know whether the sequence is based on highly redundant reads coverage, as it is for our Bactopia processed genomes used here. The final group of factors concerns choices about bioinformatic software and what parameters to use. Out of a wide range of open source options available, we chose to use highly cited tools Bakta (51) [which uses the Prodigal (52) gene finder] for annotation and PIRATE (15) for pangenome estimation. PIRATE iteratively increases the threshold to report the maximum identity that clusters each gene family and therefore avoids over-splitting gene families. PIRATE also identifies alleles within families without creating artificial paralog gene families. Tools that split paralogs into separate gene families [e.g., ROARY (53) using default parameters] will also produce larger numbers of gene families and fewer core genes. The choice of minimum

threshold for clustering proteins or genes (usually based on percentage identity of a pairwise alignment) is important. We realized from constructing the pangenome with a minimum 50% threshold that 85% of *S. aureus* gene families were clustered with at least the 90% identity. When we tested the 740-set pangenome with the minimum threshold increased to 90%, we found a similar number of core genes (2,139 at 50% minimum vs 2,085 at 90% minimum), but the number of non-core genes increased to from 3,426 to 5,472 (90%). This was because many intermediate gene families had been split at the higher threshold. However, the different threshold did not affect the key result of this study, which was that intermediate genes could be placed into two groups based on segregation with the strains defined by ANI using the $F_{ST}$ statistic. Although we did not thoroughly explore different options in this study, pangenome estimation in *S. aureus* could be further optimized in future benchmarking studies based on the genome data collected here.

We defined two classes of *S. aureus* intermediate frequency genes. Strain-diffuse genes are maintained in the population yet have a high turnover; i.e., they are gained and lost frequently (e.g., LukFS, TSST, SEA, and SEB in Fig. 4). These genes are associated with mobile elements on the chromosome, such as prophages, SaPIs, and SCCmec, and are also often found on contigs unlinked to core genes, as would be expected of plasmids. *S. aureus* strain-diffuse genes are strikingly promiscuous in their strain background (Fig. S4). This suggests high rates of horizontal transfer and, over the longer term, relatively weak barriers to genetic exchange compared to the strength of selection for strain-diffuse genes. The second class, strain-concentrated genes, segregated closely with the strain core gene background. Many of the genes cluster in the *S. aureus* genome islands, particularly vSaβ. The elements have been described as having complex, strain-specific genetic structure (54, 55). Strain-concentrated genes also include significant virulence-related functions located outside of previously defined genetic elements such as certain type VII secretion and capsule genes. Strain-concentrated genes have many fewer predicted gene gains and losses than strain-diffuse genes (Fig. 5) and a much stronger phylogenetic signal (Fig. S4). This suggests that the rate of horizontal transfer of strain-diffuse genes is much higher, and the probable reason is that they are on self-transmissible elements such as phages and plasmids (conjugative and mobilizable). The genome islands appear to have evolved from prophage or SaPIs that have acquired null mutations in their genes for site-specific recombination. We propose the mechanism of horizontal transfer of strain-diffuse genes is indirect: homologous recombination following introduction of DNA into the donor cell.

This study raises two questions about the manner in which the *S. aureus* genome evolves and the underlying selective pressures that drive the observed patterns: (i) what are the forces that create the "valley" of ANI in the range of 99.1%–99.5% (Fig. 1)? and (ii) what are the functional implications of the partitioning of intermediate genes in strain-concentrated and strain-diffuse groups? The ANI valley implies that there is a limited time that strains can survive as coherent taxonomic units, as measured by accumulation of neutral mutations. Possibly, strains are replaced from within by the wavelike expansion of successful clones. Something like this process may be happening with the expansion of USA300 since the late 1980s, gradually becoming the most common CC8 strain in the USA (56, 57). This explanation implies that strains occupy distinct niches, with adaptation possibly defined by the composition of their non-core genes (58, 59). Substrains would then be competing with each other to occupy the strain niche. New strains can also emerge from outside by genome-scale recombination events, exemplified by CC239 strains (21, 22). Judging by the relatively small size of the "99.1%–99.5% bump" (Fig. 1), these types of events may be a rare but ongoing process.

The second question we highlight concerns the functional implications of the partition of strain-concentrated and strain-diffuse genes. There is a bias for deletion in bacterial genomes (60) that implies genes maintained over time are under enduring strong selection. Conversely, the strain-diffuse gene pattern can be seen as cycles of gene gain under neutral selection (i.e., driven by gene transfer alone) or short-term

positive selection followed by rapid removal. However, we do not know of any studies that address the underlying reasons for the difference in strain-level vs substrain-level selection. Toxins are interesting in this regard because of their importance for *S. aureus* virulence. Why are some toxins maintained as core functions [e.g., alpha-toxin (*hly*)], some strain concentrated [e.g., enterotoxin G (*seg*)], and some strain diffuse, present in diverse substrains [e.g., Panton-Valentine leukocidin (*lukFS*)]? (Fig. 4). The superantigen-type toxins are split between strain-concentrated and strain-diffuse genes, suggesting that former functions may be strongly linked to strain niches. This also opens up the possibility of using the strain-concentrated genes as markers for strain identification in epidemiological studies as suggested by others (61, 62) or in metagenomic samples.

In summary, this work revealed a new partition in the structure of the *S. aureus* pangenome that will spur further studies on genome evolution and subspeciation in the species. The methodology for refining large amounts of public data, defining strains using ANI, and following strain specificity of the pangenome using $F_{ST}$ can also be applied to other bacterial species. Comparisons to other species, particularly from the *Staphylococcus* genus, will reveal the commonalities and unique selective pressures acting on the pangenome of this dangerous pathogen.

## MATERIALS AND METHODS

### Public genome collection, processing, and filtering

Bactopia v.1.7.0 was used to download and process all genomes used in this data set. Bactopia is a software pipeline for comprehensive analysis of bacterial genomes based on Nextflow (63, 64). The command *"bactopia search "Staphylococcus aureus" --prefix saureus"* was used to download all *S. aureus* short-read sequences available on Sequence Read Archive in September 2022. Bactopia used SKESA to assemble genomes, Bakta to annotate, and Snippy for variant calling (65, 66). Assembly quality was evaluated using QUAST and CheckM (67, 68). *S. aureus* CC and ST were based on the PubMLST database (18) (https://pubmlst.org/bigsdb?db=pubmlst_saureus_seq-def&page=downloadProfiles&scheme_id=1). AgrVATE v.1.0.5 was used to assign *agr* types (20). Only samples having greater than 50× coverage, mean per-read quality greater than 20, mean read length greater than 75 bp, and an assembly with less than 200 contigs were considered for the analysis (corresponding to "gold" and "silver" ranks as designated by Bactopia. Samples that were detected as not *S. aureus* according to kmer-based identification or CheckM were then removed. Coverage for all samples were capped at 100×. For every sample, bactopia performs variant calling using Snippy against an auto-chosen reference sequence based on the smallest Mash distance to a complete *S. aureus* genome in RefSeq (65, 69). For each variant identified, the allele frequencies were calculated from the bam files using bcftools mpileup (70). Samples having average minor allele frequency of >0.05 were considered mixed strains and were therefore removed. Samples having a total number of variants of >150,000 compared to the auto-chosen reference (or more than 5% of the genome) were also considered non-*S. aureus* and were removed (71). This process reduced 83,383 samples to 56,771. Since Bactopia collected and processed only short-read *S. aureus* data, we added complete *S. aureus* genome sequences to this set. Out of 1,475 complete genomes publicly available as of February 2023, 1,263 did not have any 'N' characters in their assemblies and were added to the filtered data set of 56,771, leading to a total of 58,034 genomes (56,771 short-read genomes + 1,263 complete genomes). The 212 complete genomes containing "N" characters were not used in this study.

### Substrain dereplication

Samples were grouped by their MLST types as assigned by Bactopia and for each ST, an all-vs-all Mash distance estimation (69) was run. Samples with a Mash distance of <0.0005 [approximately 50 single nucleotide polymorphisms (SNPs)] (12, 20, 72) were grouped

into clusters. A randomly chosen representative of each of these 7,954 substrains was selected for downstream analysis. Where possible, we used complete genomes as the cluster representative. Samples with unassigned STs were grouped together and treated the same. The resulting final dereplicated set of 7,954 genomes was used for pangenome construction. The representative substrains came from 1,706 MLSTs, with 386 substrains not belonging to a previously assigned ST. The uneven distribution of genomes across substrains and STs reflected the sampling skew toward well-known *S. aureus* strains from predominantly clinical settings. We found that the 15 substrains that represented the most collapsed genomes comprised 50% of the shotgun data sets. The most numerous substrain, from CC22, comprised 7,688 of the 58,034 whole genomes (13%), while there were 5,597 substrains represented by only one genome. Out of 7,954 substrains, 3,857 (48%) were in the 10 most abundant STs (ST5, ST8, ST30, ST398, ST45, ST1, ST22, ST15, ST59, and ST239), representing 39,366 out of 56,771 genomes (69%).

## Pangenome analysis

The Bakta annotation produced by the original Bactopia run was used as input for pangenome estimation with PIRATE v.1.0.5 (15). PIRATE was run using default parameters with the additional flags -a to obtain core genome alignments and -k "--diamond" to use DIAMOND for the amino acid sequence comparisons (73). SNP-sites v.2.5.1 (74) was run on the PIRATE core genome alignment to extract only polymorphic sites (709,911 sites), and the resulting alignment was used to construct a core genome phylogeny with FastTree v.2.1.11 (75) (GTR model; 1,000 bootstrap resamples). The phylogeny was visualized using the R package ggtree (76, 77). A ST93 strain (accession number GCA_000144955.2) was drawn at the root as described in (20). We used HomoplasyFinder (26) to count the number of state changes of each non-core gene on the phylogeny. geNomad v.1.5 (27) was used to predict mobile genetic elements.

## Strain definition based on ANI

All-vs-all pairwise ANI was calculated for the 7,954 dereplicated genomes using fastANI v.1.33 (71). We also calculated all-vs-all pairwise SNP distances based on the concatenated nucleotide sequences of the core genes (2,101,692 nt) using snp-dists v.0.7.0 (https://github.com/tseemann/snp-dists) and observed a similar three-peak distribution as in Fig. 2A. Strain assignments were performed based on average linkage hierarchical clustering, and samples that had an ANI of 99.5% or greater were clustered together, and this 99.5% ANI threshold also corresponded to a valley after the first peak in the SNP distance distribution (Fig. S8). We decided to use the ANI threshold based on assemblies rather than the core gene SNP threshold because (i) it is significantly faster to perform ANI comparisons, thereby making it easier to incorporate new genomes in the future; and (ii) there is existing literature corroborating the 99.5% ANI threshold (17). The average ANI of each genome with every other genome in a given cluster was calculated, and the genome with the highest average ANI was assigned as the strain representative.

## Calculating $F_{ST}$

We created a custom R function to calculate the $F_{ST}$ for each gene, with group membership defined as strain type, clonal complex, or *agr* group, depending on the purpose of the comparison. The input was a binary presence/absence data frame, with genes as columns and genomes as rows. $F_{ST}$ was calculated using Weir's formula (23).

## Creating the 740-set and 740-set-90 pangenomes

We randomly subsampled 20 substrains each from all strains with >20 substrains (37 strains). We reran PIRATE v.1.0.5 with default parameters and created a core genometree using FastTree v.2.1.11 as described above. To create the 740-set-90 pangenome, we ran the 740 genomes through PIRATE v.1.0.5 with minimum clustering threshold of 90% amino acid identity.

## Chromosomal locations of non-core genes

We used two methods for mapping chromosomal locations of non-core genes based on the co-ords output of the PIRATE v.1.0.5 pipeline for the 7954-set and 740-set pangenomes. First, we screened 377 complete substrain genomes that had *dnaA* as their first gene by BLAST and collated the start coordinate of each non-core gene. The second method was to collate the start coordinate of the nearest core gene on the same contig as each non-core gene. For each class of non-core gene, 20,000 random genes were selected as well as a control of 20,000 genes of all classes (including core). If the non-core gene was on a contig that did not have a core gene, then its status was returned as "unlinked."

## Antibiotic resistance, virulence, and phage defense functions

To assign antibiotic resistance genes, we queried representative protein sequences of each gene family of the 7954-set produced by PIRATE against the AMRFinder+ (30) database using tblastn (78) with a threshold of ≥90% identity as a match. We filtered out the virulence-associated genes using matches for the terms "serine_protease," "enterotoxin," "hemolysin," "Panton," "adhesin," "complement," "aureolysin," "exfoliative," "toxin," "intracellular_survival," "serum_survival," and "leukocidin" and kept the remainder as antibiotic resistance gene matches. To assign phage defense-related functions, we queried the 7954-set representative proteins against the online defensefinder database (32) (https://defense-finder.mdmparis-lab.com/) on 17 October 2023.

## Statistical analysis and data visualization

All statistics and t-SNEs were performed in R using the package rstatix (79). All plots were visualized using R package ggplot2 (80). Other visualizations were performed using draw.io and Sakneymatic (81, 82).

## ACKNOWLEDGMENTS

D.B.W. and T.D.R. were supported by an Emory University Synergy II_Nexus/P3 award. T.D.R. was supported by funding from National Institutes of Health (NIH) awards AI158452 and AI139188. V.R. was supported by NIH AI139188 and the NIH T32 AI138952 award (Infectious Disease Across Scales Training Program). D.B.W. was supported by funding from the Simons Foundation (Mathematical Modeling of Living Systems Investigator award 508600), the Sloan Foundation (Research Fellowship FG-2021–16667), and the NSF (award 2146260).

We thank Megan Phillips and Anayancy Ramos Facio for discussions about the manuscript.

## AUTHOR AFFILIATIONS

[1]Microbiology and Molecular Genetics Program, Graduate Division of Biological and Biomedical Sciences, Laney Graduate School, Emory University, Atlanta, Georgia, USA
[2]Division of Infectious Diseases, Department of Medicine, Emory University, Atlanta, Georgia, USA
[3]Department of Physics, Emory University, Atlanta, Georgia, USA

## PRESENT ADDRESS

Vishnu Raghuram, Department of Clinical Microbiology, Umeå University, Umeå, Sweden
Robert A. Petit III, Wyoming Public Health Laboratory, Cheyenne, Wyoming, USA
Rohan Mehta, Department of Biology, Elmhurst University, Elmhurst, Illinois, USA

## AUTHOR ORCIDs

Vishnu Raghuram  http://orcid.org/0000-0002-7435-6435

Timothy D. Read  http://orcid.org/0000-0001-8966-9680

## FUNDING

| Funder | Grant(s) | Author(s) |
|---|---|---|
| HHS \| National Institutes of Health (NIH) | AI139188 | Vishnu Raghuram |
| | | Timothy D. Read |
| HHS \| National Institutes of Health (NIH) | AI138952 | Vishnu Raghuram |
| HHS \| National Institutes of Health (NIH) | AI158452 | Timothy D. Read |
| Simons Foundation (SF) | 508600 | Daniel B. Weissman |
| Alfred P. Sloan Foundation (APSF) | FG-2021-16667 | Daniel B. Weissman |
| National Science Foundation (NSF) | 2146260 | Daniel B. Weissman |

## DATA AVAILABILITY

PIRATE pangenome outputs, genes and strain lists, and representative genome sets are available on Zenodo (https://zenodo.org/records/10471309). R code for generating figures can be found in https://github.com/VishnuRaghuram94/SCAPE.

## ADDITIONAL FILES

The following material is available online.

### Supplemental Material

**Supplemental figures and table (mSystems00143-24-S0001.docx).** Figures S1 to S8 and Table S1.

### Open Peer Review

**PEER REVIEW HISTORY (review-history.pdf).** An accounting of the reviewer comments and feedback.

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
