## [Reviewer comments · mSystems]

Average Nucleotide Identity based *Staphylococcus aureus* strain grouping allows identification of strain-specific genes in the pangenome

Vishnu Raghuram, Robert Petit III, Zach Karol, Rohan Mehta, Daniel Weissman, and Timothy Read

Corresponding Author(s): Timothy Read, Emory University School of Medicine

Review Timeline:

Submission Date:	January 29, 2024
Editorial Decision:	February 29, 2024
Revision Received:	April 9, 2024
Accepted:	April 16, 2024

Editor: Juliette Hayer

Reviewer(s): The reviewers have opted to remain anonymous.

Transaction Report:

DOI: <https://doi.org/10.1128/msystems.00143-24>

Re: mSystems00143-24 (Average Nucleotide Identity based *Staphylococcus aureus* strain grouping allows identification of strain-specific genes in the pangenome)

Dear Dr. Timothy D Read:

I am pleased to announce that your manuscript has been accepted for publication in mSystems, with minor modifications. Please take into account the reviewers valuable comments, I think the manuscript will benefit from addressing the questions they raised.

Revision Guidelines

Sincerely,
Juliette Hayer
Editor
mSystems

Reviewer #1 (Comments for the Author):

Summary

Employing one of the largest and most diverse genome datasets for *S. aureus*, the authors conduct an ANI-based strain typing

within the species. Importantly, the database was well curated not only for quality but also for redundancy. The authors go beyond the strain definition and analyse the frequency distribution of the accessory gene families taking into account whether they fall within and/or between strains. This is a solid and well-conducted study that adds interesting insights to the burgeoning field of genomic definition/delimitation of intra-species units. However, some aspects can improve the paper; see major and minor comments below.

Major comments

1. My most salient comment has to do with the fact that the relationship between the ANI-based strains and ST was not mentioned in the article. ST assignment is far more common than CC designation for many bacterial species. Furthermore, this will give another granularity level to the analysis, as the ST assignment is below the CC assignment. Thus, including the ST level will not only increase the level of detail but will also help to compare the patterns of this study with future and previous studies.

2. One interesting finding is that strain-concentrated genes, which are part of the accessory genome, have phylogenetic signal, thus these could be phylogenomic markers for genomic epidemiology studies. The authors might want to comment on this, especially in the context of recent discussions about using the accessory genome in addition to the core genome to conduct genomic epidemiology. See refs below.

<https://pubmed.ncbi.nlm.nih.gov/35544058/>

<https://pubmed.ncbi.nlm.nih.gov/34282943/>

C) Considering future strain assignment, lines 370-372 in the manuscript, please provide a supplementary table listing the 145 representative genomes with their metadata (ST, CC, host, etc.). This will be very helpful for future studies that want to conduct strain assignments.

Minor comments

This is more a suggestion than anything else, instead of "non-core genome/genes" the authors could use "accessory genome/genes", which is considerably more frequently used.

Lines 103-105: please state how many genes were left after removing redundancy. I know this is stated in the methods but given the structure of the article (Intro, Results, Discussion and Methods) this will make it easy on the reader.

Line 121 and some other lines in the text: I'd suggest the authors use "homologous groups/genes" instead of "orthologous genes". By definition, orthologous genes are those reflecting the species tree - genes in different species that evolved from a common ancestral gene by speciation- and are bound to be very few in any given bacterial species.

Lines 188-189: How did you come up with the Fst threshold of 0.75? Given Figure 3A, 0.625 seems like a better option.

Line 290: What do the authors mean by "orthologous methods"? I guess "methods" is just fine.

Line 604: What do the authors mean by "core pangnome tree"? I guess you meant to say "a core genome tree", didn't you?

Reviewer #2 (Comments for the Author):

In this manuscript, the authors analyze all publicly available *Staphylococcus aureus* genomes to characterize accessory gene content, or "non-core" genes, between distinct strains. Importantly, strains were operationally defined as sharing an average nucleotide identity of >99.5% based on the most stringent possible natural cut-off of core genome comparison. Thus, the definition of strain here is more stringent than standard MLST or ST designations. The authors found 145 distinct strains. Non-core genes were then divided into two main categories (constrained vs. diffuse) based on how restricted their presence is to a specific strain, again using a natural cut-off observed in the dataset based on the F(ST) metric. The authors found that 'strain-constrained' accessory genes differ from 'strain-diffuse' accessory genes in the types of functions encoded, locations within genomes, and association with mobile genetic elements. In effect, the authors have operationally defined the core and non-core genomes of each and all *S. aureus* strains separately, using logical cut-offs from high resolution sequencing data. These data provide a foundation for understanding the dynamics of gene flow in *S. aureus*. Although the work is largely descriptive, it is complete, logical, and rigorous, making it an important contribution both in the methodologies employed and in the precision of the main findings. Thus, my comments are largely focused on the writing and presentation of the manuscript, which could be improved.

Major:

1. In general, the Results and Discussion sections are too long and/or somewhat redundant with each other. The writing could

be more focused, which would improve both clarity and impact. See specific comments below.

2. Results: Some parts are too long and could be moved to Methods or Supplemental Data.

a. Lines 97-106: Move the details to Methods and consider something more brief, such as a single sentence similar to below:

To obtain a comprehensive view of *S. aureus* genetic diversity, we first examined all 83,383 whole genome data sets available on the NCBI website as of September 2022 for data quality and redundancy, and selected 7,954 high quality "substrains" that represented the overall diversity (Figure 1, Figure S1A; Methods).

b. Lines 108-117: Move details to Supplemental Data and consider something more brief, such as a single sentence similar to below:

As expected, the distribution of substrains and STs reflected the sampling bias of *S. aureus* from clinical settings (Table SX).

c. Lines 141-170. Please just briefly summarize and reference supplemental file or table for the details. For example, I think just mentioning that "7 out of 10 clonal complexes were split into multiple strains (Figure SX)" is sufficient. We already know this would happen because 145 is a larger number than 10. These exact parsing details aren't important to the narrative of this manuscript.

3. Lines 251-253: "Together these results showed that strain-concentrated genes provided more information about gene content differences between strains than other non-core genes." Isn't this statement self-evident? By definition, won't strain-concentrated genes better discern gene content differences between strains than more strain-diffuse genes? This statement is issued as a logical conclusion of the analysis of Figure 6, but it's a statement that just always has to be true, right? I'm not sure you actually need any empirical data at all to make this statement. Your data nicely characterize this idea and assign genes to different categories (concentrated vs diffuse), but they don't 'prove' something that is inherently just true. Consider re-writing this section to emphasize you are simply exploring different ways to visualize this concept. I think it's a bit misleading to state you that you "showed" evidence that strain-concentrated genes are, well, specific to certain strains. Its circular logic, is it not? Unless I am confused about something?

4. Discussion: It is too long. Much of the discussion is redundant with the Results or unfocused. The Discussion should focus on comparison to literature, new findings to the field, and drive home a few main take-away points.

a. Lines 347-355. Redundant with Results section or self-evident. Consider cutting.

b. Lines 357-368. Please cut this. Not helpful in this context of a primary Research Article. You did a nice job of defining what you meant by "strain" in the Results. The semantics of this issue can be discussed elsewhere I think.

c. Lines 386-432. Please condense. Could probably be half the length.

d. Lines 434-463. Please condense. Can you make this just one paragraph about HGT?

e. Lines 465-480. Consider cutting the rare genes discussion completely. It doesn't add much. It is addressed in the Results and is not a well-defined category. Certainly not a take-home point.

f. Lines 482-524. I'm not sure these questions are specific to or were revealed by this study. Aren't these long-standing questions in evolutionary biology? Consider cutting or condensing and just stating how you think your specific data will help here. Please be brief.

Minor:

1. Line 35 (Abstract): Change "antibiotic genes" to 'antibiotic resistance genes'.

2. Lines 95-96: Remove "from a large public genome dataset" from the subsection title. The reader already knows the data source.

3. Line 99: Change "...website in September 2022" to '...website as of September 2022'.

4. Line 129: By definition, distinct peaks will be separated by distinct valleys. Consider changing "...histogram there was a clear pattern of three strong peaks separated by distinct valleys (Figure 2A)" to '...histogram, we observed three major ANI peaks (Figure 2A).'

5. Line 136: Change "...marked each with a suffix "99.5_"." to 'marked each with a suffix "99.5_i", where i denotes the unique strain number of one of the 145 strains.'

6. Line 203: "...were either almost entirely present or absent in each strain background." Can you give this added context in quantitative terms? The F(ST) value of >0.9 is given and we have Figure 4 to look at, so we already have a general sense, but it would be nice to know what "almost entirely present or absent" means in absolute numbers and/or percentages. Could say something like 'for example, strain-specific presence or absence of LukDE was >X% conserved within each strain.'

7. Line 208: Can you better explain the motivation for creating the "740-set" before discussing the results of this analysis? Perhaps better define what you mean by 'balance' and what the potential issue with an 'unbalanced' dataset is first. This way the reader knows why you are doing this. Eventually, the last sentence of this paragraph states you don't see these signals with the main dataset, so presumably the motivation is that an unbalanced dataset can obscure the signal that you are interested in uncovering. I think it would read better if it was a more logical flow. I was able to understand this eventually, but it took some effort. Consider first telling the reader that you were looking for this specific signal in the main dataset, but didn't see it, so wondered if this was due to sampling imbalance among strains. Therefore, you created the 740-set... and found that yes, the signal does in fact emerge... in other words, lay out the thought process chronologically to help the reader along. I also wonder if

using the 4-panel figure of Figure S6 would be more appropriate here to help carry the logic, so that the reader can see and compare the balanced and unbalanced datasets side by side.

8. Line 211-214: "The 740-set had similar numbers of core and intermediate genes... The F(ST) distribution of the 740-set to the original pangenome was almost identical." What does "almost identical" mean? Please be quantitative. Considering showing an overlay of the distributions in Figure S2 or just adding the 740-set distributions as new panels in Figure S2.

9. Line 221-222: "From Figure 5A it is clear that rare gene distributions were extensions of the trends seen in intermediate genes." This sentence is vague and I'm not exactly sure what you are trying to convey here. I think you are trying to say that a fraction of the rare genes appear to follow the strain-diffuse pattern in Figure 5A because the rare and diffuse data points are intermixed, separate from the concentrated genes, and hugging the theoretical 'random distribution' line, suggesting that many of the genes originally classified as 'rare' are actually prevalent enough to be shown to be diffusely distributed among the strains included in this analysis. Can you edit this sentence so it is more precise? [Note: Above, I suggested cutting all discussion of rare genes from the Discussion, so if you want to emphasize the "meaningless" or inability to assign concentrated/diffuse categories to 'very rare genes' simply because they are so rare, do it briefly in the Results here. Although, this is also sort of self-evident, so not sure you really need to mention it.]

10. Line 253: Consider a new paragraph break here.

11. Figure S2: what are the bin sizes in each histogram? Please add this info to the legend.

12. Figure S5: please explain in the legend what the box plots represent (mean, median, error?) and also define HFSTI and LFSTI.

13. Do you think it would be helpful for the field to have a complete, well annotated genome from each of the 145 strains on NCBI? Or at least the 37 most prevalent strains? How many of the 37 (or 145) strains are even represented by a complete genome on NCBI currently? Perhaps this should be a goal. Curious what you think. I'm not saying you need to add this to the text, unless you think so.

14. Do you think it would be helpful for the field to know the set of highly strain-concentrated genes present in each of the 37 most prevalent strains? Can we think of this category of 'non-core' genes as strain-specific 'core accessory genes'? Is it worth supplying these gene name lists as Supplemental Data? Curious what you think. If you think useful, then consider adding this to the supplement.

RESPONSE TO REVIEWERS

mSystems00143-24R1 - Average Nucleotide Identity based *Staphylococcus aureus* strain grouping allows identification of strain-specific genes in the pangenome

We thank the reviewers for their constructive feedback. Please find our responses underneath each of the reviewer's comments. Our responses are in red. The line numbers in the response correspond to the line numbers in the revised version.

Vishnu Raghuram and Tim Read , on behalf of all the authors

Reviewer #1 (Comments for the Author):

Summary

Employing one of the largest and most diverse genome datasets for *S. aureus*, the authors conduct an ANI-based strain typing within the species. Importantly, the database was well curated not only for quality but also for redundancy. The authors go beyond the strain definition and analyse the frequency distribution of the accessory gene families taking into account whether they fall within and/or between strains. This is a solid and well-conducted study that adds interesting insights to the burgeoning field of genomic definition/delimitation of intra-species units. However, some aspects can improve the paper; see major and minor comments below.

Major comments

1. My most salient comment has to do with the fact that the relationship between the ANI-based strains and ST was not mentioned in the article. ST assignment is far more common than CC designation for many bacterial species. Furthermore, this will give another granularity level to the analysis, as the ST assignment is below the CC assignment. Thus, including the ST level will not only increase the level of detail but will also help to compare the patterns of this study with future and previous studies.

Based on the phylogeny, and our ANI based screening, in general CCs are quite well aligned with the natural strains of *S. aureus*. Moreover, ~70% of sequenced *S. aureus* genomes (as of this study) belong to only 10 STs out of 1706, meaning most discovered STs are very rare with very few representatives. With this in mind we thought STs would simply add artificial granularity to the analysis and not give us any more information

about the population structure than CCs already do.

We added a sentence in line 137-139 stating how many times a ST was found in multiple strains

2. One interesting finding is that strain-concentrated genes, which are part of the accessory genome, have phylogenetic signal, thus these could be phylogenomic markers for genomic epidemiology studies. The authors might want to comment on this, especially in the context of recent discussions about using the accessory genome in addition to the core genome to conduct genomic epidemiology. See refs below.

<https://pubmed.ncbi.nlm.nih.gov/35544058/>

<https://pubmed.ncbi.nlm.nih.gov/34282943/>

Added a sentence to line 433 - 435 and included these references.

3. Considering future strain assignment, lines 370-372 in the manuscript, please provide a supplementary table listing the 145 representative genomes with their metadata (ST, CC, host, etc.). This will be very helpful for future studies that want to conduct strain assignments.

We have a zenodo repository (<https://zenodo.org/records/10471309>) that has this table (isolate_substrain_strain_freq_st_CC.tsv) along with several other files (representative assemblies, pangenome outputs, alignments, gene trees etc). This is stated in the ‘Data availability’ section. We do not have host information as it was not relevant to this study. Moreover, from our previous analyses we found only ~25% of publicly available genomes to have reliable metadata (<https://journals.asm.org/doi/10.1128/spectrum.01334-21>)

Minor comments

This is more a suggestion than anything else, instead of "non-core genome/genes" the authors could use "accessory genome/genes", which is considerably more frequently used.

We understand that “accessory” may be more commonly used but at this stage of the manuscript we believe it does not change the meaning or the conclusions, and could possibly introduce errors.

Lines 103-105: please state how many genes were left after removing redundancy. I know this is stated in the methods but given the structure of the article (Intro, Results, Discussion and Methods) this will make it easy on the reader.

This is stated in line 95 in the Results section

Line 121 and some other lines in the text: I'd suggest the authors use "homologous groups/genes" instead of "orthologous genes". By definition, orthologous genes are those reflecting the species tree - genes in different species that evolved from a common ancestral gene by speciation- and are bound to be very few in any given bacterial species.

We used the same terminology used in the original PIRATE publication. To avoid confusion, we have removed any mention of the word "orthologous" and simply refer to them as gene families

Lines 188-189: How did you come up with the F_{st} threshold of 0.75? Given Figure 3A, 0.625 seems like a better option.

We chose 0.75 as it was a more stringent threshold plus the tSNE from the 740-set (Fig 6) shows that the 0.75 cutoff resolved strains into distinct clusters. Ultimately, this threshold is subjective and more work can be done to empirically assess the association of each gene to each strain background.

Line 290: What do the authors mean by "orthologous methods"? I guess "methods" is just fine.

We changed it to alternate methods

Line 604: What do the authors mean by "core pangnome tree"? I guess you meant to say "a core genome tree", didn't you?^[SEP]

Correct, fixed the typo.

Reviewer #2 (Comments for the Author):

In this manuscript, the authors analyze all publicly available *Staphylococcus aureus* genomes to characterize accessory gene content, or "non-core" genes, between distinct

strains. Importantly, strains were operationally defined as sharing an average nucleotide identity of >99.5% based on the most stringent possible natural cut-off of core genome comparison. Thus, the definition of strain here is more stringent than standard MLST or ST designations. The authors found 145 distinct strains. Non-core genes were then divided into two main categories (constrained vs. diffuse) based on how restricted their presence is to a specific strain, again using a natural cut-off observed in the dataset based on the F(ST) metric. The authors found that 'strain-constrained' accessory genes differ from 'strain-diffuse' accessory genes in the types of functions encoded, locations within genomes, and association with mobile genetic elements. In effect, the authors have operationally defined the core and non-core genomes of each and all *S. aureus* strains separately, using logical cut-offs from high resolution sequencing data. These data provide a foundation for understanding the dynamics of gene flow in *S. aureus*. Although the work is largely descriptive, it is complete, logical, and rigorous, making it an important contribution both in the methodologies employed and in the precision of the main findings. Thus, my comments are largely focused on the writing and presentation of the manuscript, which could be improved.

Major:

1. In general, the Results and Discussion sections are too long and/or somewhat redundant with each other. The writing could be more focused, which would improve both clarity and impact. See specific comments below.

2. Results: Some parts are too long and could be moved to Methods or Supplemental Data.

a. Lines 97-106: Move the details to Methods and consider something more brief, such as a single sentence similar to below:

To obtain a comprehensive view of *S. aureus* genetic diversity, we first examined all 83,383 whole genome data sets available on the NCBI website as of September 2022 for data quality and redundancy, and selected 7,954 high quality "substrains" that represented the overall diversity (Figure 1, Figure S1A; Methods).

We followed the reviewer's suggestion here (Lines 87 - 91).

b. Lines 108-117: Move details to Supplemental Data and consider something more brief, such as a single sentence similar to below:

As expected, the distribution of substrains and STs reflected the sampling bias of *S. aureus* from clinical settings (Table SX).

We moved this section of the results into the methods (491 - 500).

c. Lines 141-170. Please just briefly summarize and reference supplemental file or table for the details. For example, I think just mentioning that "7 out of 10 clonal complexes were split into multiple strains (Figure SX)" is sufficient. We already know this would happen because 145 is a larger number than 10. These exact parsing details aren't important to the narrative of this manuscript.

We have reduced this section

3. Lines 251-253: "Together these results showed that strain-concentrated genes provided more information about gene content differences between strains than other non-core genes." Isn't this statement self-evident? By definition, won't strain-concentrated genes better discern gene content differences between strains than more strain-diffuse genes? This statement is issued as a logical conclusion of the analysis of Figure 6, but it's a statement that just always has to be true, right? I'm not sure you actually need any empirical data at all to make this statement. Your data nicely characterize this idea and assign genes to different categories (concentrated vs diffuse), but they don't 'prove' something that is inherently just true. Consider re-writing this section to emphasize you are simply exploring different ways to visualize this concept. I think it's a bit misleading to state you that you "showed" evidence that strain-concentrated genes are, well, specific to certain strains. Its circular logic, is it not? Unless I am confused about something?

We have deleted the redundant sentence.

4. Discussion: It is too long. Much of the discussion is redundant with the Results or unfocused. The Discussion should focus on comparison to literature, new findings to the field, and drive home a few main take-away points.

a. Lines 347-355. Redundant with Results section or self-evident. Consider cutting.

We would prefer to leave this section in the paper. It is a matter of taste but we think a

brief recap of major results at the top of the discussion is helpful for the reader

b. Lines 357-368. Please cut this. Not helpful in this context of a primary Research Article. You did a nice job of defining what you meant by "strain" in the Results. The semantics of this issue can be discussed elsewhere I think.

OK - this section has been deleted.

c. Lines 386-432. Please condense. Could probably be half the length.

This para has been condensed by removing two sentences

d. Lines 434-463. Please condense. Can you make this just one paragraph about HGT?

We have condensed these two paragraphs into one and shortened.

e. Lines 465-480. Consider cutting the rare genes discussion completely. It doesn't add much. It is addressed in the Results and is not a well-defined category. Certainly not a take-home point.

This paragraph has been deleted

f. Lines 482-524. I'm not sure these questions are specific to or were revealed by this study. Aren't these long-standing questions in evolutionary biology? Consider cutting or condensing and just stating how you think your specific data will help here. Please be brief.

We have deleted some sentences to shorten these paragraphs. We disagree with the reviewer on the point that these are not questions raised specifically by our data. The two questions we focus on- what is the cause of ANI 99.5% threshold for strains, and the partition of certain functions in strain-diffuse and strain-concentrated bin are key topics that stem from our results that should be examined more in the future.

Minor:

1. Line 35 (Abstract): Change "antibiotic genes" to 'antibiotic resistance genes'.

Done

2. Lines 95-96: Remove "from a large public genome dataset" from the subsection title. The reader already knows the data source.

Done

3. Line 99: Change "...website in September 2022" to '...website as of September 2022'.

Done (we presume the reviewer meant to change "as of" to "in")

4. Line 129: By definition, distinct peaks will be separated by distinct valleys. Consider changing "...histogram there was a clear pattern of three strong peaks separated by distinct valleys (Figure 2A)" to '...histogram, we observed three major ANI peaks (Figure 2A).'

Done

5. Line 136: Change "...marked each with a suffix "99.5_"." to 'marked each with a suffix "99.5_i", where i denotes the unique strain number of one of the 145 strains.'

Done

6. Line 203: "...were either almost entirely present or absent in each strain background." Can you give this added context in quantitative terms? The F_{ST} value of >0.9 is given and we have Figure 4 to look at, so we already have a general sense, but it would be nice to know what "almost entirely present or absent" means in absolute numbers and/or percentages. Could say something like 'for example, strain-specific presence or absence of LukDE was $>X\%$ conserved within each strain.'

Added this sentence to provide an example "For example LukD was not present in 60/145 (41%) strains but present in $> 80\%$ of 77/45 (53%) strains." (Line 171 - 173)

7. Line 208: Can you better explain the motivation for creating the "740-set" before discussing the results of this analysis? Perhaps better define what you mean by 'balance'

and what the potential issue with an 'unbalanced' dataset is first. This way the reader knows why you are doing this. Eventually, the last sentence of this paragraph states you don't see these signals with the main dataset, so presumably the motivation is that an unbalanced dataset can obscure the signal that you are interested in uncovering. I think it would read better if it was a more logical flow. I was able to understand this eventually, but it took some effort. Consider first telling the reader that you were looking for this specific signal in the main dataset, but didn't see it, so wondered if this was due to sampling imbalance among strains. Therefore, you created the 740-set... and found that yes, the signal does in fact emerge... in other words, lay out the thought process chronologically to help the reader along. I also wonder if using the 4-panel figure of Figure S6 would be more appropriate here to help carry the logic, so that the reader can see and compare the balanced and unbalanced datasets side by side.

We recognize that we need to justify this more carefully and we have adjusted the text:

“The 7,954 representative substrains were distributed unevenly, with 58 strains having a single substrain and 15 having > 100 . This “unbalanced” sampling was an obstacle to visualizing the species gene abundance patterns as genes that were present even in a low percent of the most numerous strains would still account for more substrains than the rarest strains. To investigate the differences between strain-concentrated and strain-diffuse genes further in a *S. aureus* pangenome with more balanced sampling, we created the “740-set”, created by randomly sampling 20 shotgun assembled substrains from the most common 37 strains to make a more balanced sampling of *S. aureus*” - Lines 178 - 183

8. Line 211-214: "The 740-set had similar numbers of core and intermediate genes... The F(ST) distribution of the 740-set to the original pangenome was almost identical." What does "almost identical" mean? Please be quantitative. Considering showing an overlay of the distributions in Figure S2 or just adding the 740-set distributions as new panels in Figure S2.

We added a new panel showing this in Fig S3 (Fig S3B)

9. Line 221-222: "From Figure 5A it is clear that rare gene distributions were extensions of the trends seen in intermediate genes." This sentence is vague and I'm not exactly sure what you are trying to convey here. I think you are trying to say that a fraction of the rare genes appear to follow the strain-diffuse pattern in Figure 5A because the rare and diffuse data points are intermixed, separate from the concentrated genes, and hugging the

theoretical 'random distribution' line, suggesting that many of the genes originally classified as 'rare' are actually prevalent enough to be shown to be diffusely distributed among the strains included in this analysis. Can you edit this sentence so it is more precise? [Note: Above, I suggested cutting all discussion of rare genes from the Discussion, so if you want to emphasize the "meaningless" or inability to assign concentrated/diffuse categories to 'very rare genes' simply because they are so rare, do it briefly in the Results here. Although, this is also sort of self-evident, so not sure you really need to mention it.]

As we followed the reviewer's idea of removing the discussion of rare genes it is better to remove the last sentence of this paragraph about rare gene distribution on Figure 5A

10. Line 253: Consider a new paragraph break here.

Added a paragraph break before the sentence starting "We suspected .." - Line 222

11. Figure S2: what are the bin sizes in each histogram? Please add this info to the legend.

Done

12. Figure S5: please explain in the legend what the box plots represent (mean, median, error?) and also define HFSTI and LFSTI.

Added a description of the boxes and replaced HFSTI and LFSTI, which are terms that we no longer use with "concentrated" and "diffuse", meaning strain-concentrated and strain-diffuse.

13. Do you think it would be helpful for the field to have a complete, well annotated genome from each of the 145 strains on NCBI? Or at least the 37 most prevalent strains? How many of the 37 (or 145) strains are even represented by a complete genome on NCBI currently? Perhaps this should be a goal. Curious what you think. I'm not saying you need to add this to the text, unless you think so.

14. Do you think it would be helpful for the field to know the set of highly strain-concentrated genes present in each of the 37 most prevalent strains? Can we think of this category of 'non-core' genes as strain-specific 'core accessory genes'? Is it worth supplying these gene name lists as Supplemental Data? Curious what you think. If you think useful, then consider adding this to the supplement.

In regards to points 13 and 14: we have placed extensive data sets (~30 GB) on a Zenodo repository (<https://zenodo.org/records/10471309>) that is referenced at the end of the paper. The Zenodo data encompass all the questions raised by the reviewer (and many other possible data needs). The annotations for the 145 strains can be downloaded from here and therefore it is not necessary to upload them to NCBI, where they would be hard to access as a collective set anyway.

Re: mSystems00143-24R1 (Average Nucleotide Identity based *Staphylococcus aureus* strain grouping allows identification of strain-specific genes in the pangenome)

Dear Dr. Timothy D Read:

You have addressed all the reviewers comments and we are satisfied with the work you have done for revising the manuscript. Therefore, I am happy to announce that your manuscript is accepted for publication in mSystems. I am forwarding it to the ASM production staff for publication. Your paper will first be checked to make sure all elements meet the technical requirements. ASM staff will contact you if anything needs to be revised before copyediting and production can begin. Otherwise, you will be notified when your proofs are ready to be viewed.

Cover Image Submissions: If you would like to submit a potential Cover Image, please email a file and a short legend to msystems@asmusa.org. Please note that we can only consider images that (i) the authors created or own and (ii) have not been previously published. By submitting, you agree that the image can be used under the same terms as the published article. Image File requirements: TIF/EPS, 7.5 inches wide by 8.25 inches tall (at least 2,250 pixels wide by 2,475 pixels tall), minimum 300 dpi resolution (600 dpi preferred), RGB, and no figure elements, e.g., arrows or panel labels. The legend should be a short description of the image, 1-2 sentences recommended.

Sincerely,
Juliette Hayer

Editor
mSystems

Reviewer #1 (Comments for the Author):

The authors have addressed my concerns.

Reviewer #2 (Comments for the Author):

All of my comments were addressed. I think the manuscript has been improved.